



# A Global Lake/Reservoir Surface Extent Dataset (GLRSED): An integration of HydroLAKES, GRanD and OpenStreetMap

Bingxin Bai[1,2], Lixia Mu[1], Ge Chen[1,2], Yumin Tan[3]

[1]Department of Marine Technology, Ocean University of China, Qingdao, 266100, China
[2]Qingdao National Laboratory for Marine Science and Technology Development Center, Qingdao, 266237, China
[3]School of Transportation Science and Engineering, Beihang University, Beijing, 100191, China

*Correspondence to*: Bingxin Bai (baibingxin@ouc.edu.cn)

**Abstract.** Global lake/reservoir surface water extent is the basic input data for many studies. Although there are some datasets at present, there are problems such as incomplete or spatial inconsistency exist among them due to various reasons

like different data sources and dynamic change characteristics of the surface water. In this paper, a new Global Lake/Reservoir Surface Extent Dataset (GLRSED) that contains spatial extent and basic attributes (e.g., name, area, lake type and source) of 2.17 million lakes/reservoirs was produced based on HydroLAKES, GRanD and OpenStreetMap. In addition, by overlaying with mountain data, we identified the lakes/reservoirs located in mountain areas. By overlaying with the Global geReferenced Database of Dams (GOODD) and Georeferenced Global Dams and Reserves (GeoDAR) dataset,

we partitioned human-managed reservoirs from natural lakes. Lakes/reservoirs on the rivers were identified by overlaying with the SWOT Mission River Database (SWORD). Using the same method, we identified endorheic, glacier-fed and permafrost-fed lakes. Furthermore, the coverage of Surface Water and Ocean Topography (SWOT) ground track to each lake/reservoir in GLRSED was calculated to explore the potential of SWOT for monitoring lakes. The dataset could provide basic data for global lake/reservoir monitoring, enabling the study on the impact of human actions and climate changes on

lake/reservoir freshwater availability. The GLRSED database is available at https://doi.org/10.5281/zenodo.8121174 (Bai et al., under review, 2023).

## 1 Introduction

Lakes/reservoirs could provide support for flood control, agricultural irrigation, power generation and drinking water, which are important resources for human survival and natural ecological maintenance (Cooley et al., 2021; Verpoorter et al., 2014;

Zhao et al., 2022). The precise spatial expansion of lakes/reservoirs is often the fundamental data for many studies, such as locating the initial position of lake water surface from a large number of satellite images and reducing the image processing range for the monitoring of water surface changes (Donchyts et al., 2022). Different types of lakes/reservoirs have different characteristics, such as glacial lakes, which are often regarded as climate indicators (Adrian et al., 2009; Filazzola, 2020; Sharma et al., 2019). Therefore, the accurate spatial expansion, type and other information of lakes/reservoirs are crucial for



understanding and modeling various earth system processes, together with their interactions with the environment (Kraemer
      et al., 2020; Messager et al., 2016).

      Many datasets on lake/reservoir surface extent have been developed (Table 1 lists some). However, the lake/reservoir surface
      expansion in these datasets are not always well defined, or there is a problem of inconsistent geographical spatial boundaries
      between various datasets. The Global Reservoir and Dam Database (GRanD) (Lehner et al., 2011) was created through joint
international efforts to compile the existing data on reservoirs and dams, and gathered in one reliable database. Data
      dissimilarities and record gaps were corrected during the development of those databases. However, this database is still
      incomplete as the information about many lake/reservoirs (e.g., small and medium-sized lake/reservoirs) are missing.
      HydroLAKES (Messager et al., 2016) is an aggregation of multiple datasets and containing 1.43 million individual global
      lakes and reservoirs, which to date, is a dataset widely used by the hydrology community as one of the most comprehensive
databases of static lake polygons. Nevertheless, the spatial range of some lakes (see an example in Figure S1 in the
      supplementary material) in HydroLAKES is small or missing.. OpenStreetMap (OSM) (OpenStreetMap contributors, 2022)
      is a freely-available geodata, which has been used in a wide range of Geographic Information Systems (GIS) and
      applications as an alternative or supplementary of other authoritative datasets (Brovelli & Zamboni, 2018). As it is provided
      by multiple sources and volunteers, its attributes are mostly missing. Other datasets, such as Joint Research Centre (JRC)
Global Surface Water (GSW) (Pekel et al., 2016), are created based on three million satellite images from Landsat. The
      waterbody delineation from satellite images would be affected by atmospheric conditions (e.g., cloud) or topographic
      elements and influence the correct classification of water. Therefore, despite that the classification accuracy of GSW is high,
      there are data gaps caused by persistent cloud cover and Scan Line Corrector (SLC)-off.

      In general, the main challenge is that all lake/reservoir datasets available have certain limitations regarding their coverage
and quality. A single dataset is still incomplete, and there are also geometric differences of a feature present among these
      datasets (see Figure S1). The dynamic nature of water extends in both space and time. To avoid omissions, data representing
      a wider spatial range of water bodies is often needed in research. A full utilization of the above datasets may lead to the
      generation of lake/reservoir datasets with a better overall quality. Although, the above data have limitations, the main
      advantage is that they offer the possibility of taking short- or long-term changes of surface water into consideration by
combining images at different time.

      In this study, we integrated the spatial information of existing lake/reservoir datasets (i.e., HydroLAKES, GRanD and OSM),
      creating a more comprehensive Global Lake/Reservoir Surface Extent Dataset (GLRSED) in vector format and classifying
      the registered lakes/reservoirs into different categories (e.g., reservoirs, glacier-fed lakes and permafrost-fed lakes, etc.). We
      describe the data and methods used to develop GLRSED in Section 2 and 3. The characteristics of the resulting datasets, as
well as the implications for GLRSED and collaborative analyses with other broader datasets are described in Section 4. The
      GLRSED database could provide researchers with basic information on lake/reservoir locations, spatially explicating

inundation areas and related details. This study would arouse a positive contribution to various fields like surface water monitoring, climate changes and sustainable development goals (SDGs) monitoring.

## 2 Data

### 2.1 Datasets of Lake/reservoir Surface Extent

### 2.1.1 HydroLAKES

The HydroLAKES (Messager et al., 2016) dataset is an amalgamation of several sources, including topographic and remote sensing datasets. It contains 1.43 million individual global lakes and reservoirs that are greater than 0.1 km2, and attributes include surface area, perimeter, mean depth and volume, etc. As HydroLAKES is one of the most comprehensive and

widely-used dataset on static lake polygons to date, we will integrate more comprehensive data based on it. The HydroLAKES dataset is offered for free for scientific and educational applications at http://www.hydrosheds.org.

### 2.1.2 OpenStreetMap

OSM (OpenStreetMap contributors, 2022) is based on the collection of geographic information gathered and updated by volunteers. The data sources are from devices such as the Global Positioning System (GPS) and cadastral data, manual

digitization (editing) on medium- and high-resolution satellite as well as aerial imagery (Barron et al., 2014), which have the nature of global coverage and updateable. Several studies show that compared with authoritative datasets, OSM performs very well when it comes to positional accuracy (Haklay, 2010; Moscholaki, 2020). With these in mind, we take OSM (OpenStreetMap contributors, 2022) as a part of the integrated data. The OSM data is under the Open Data Commons Open Database License (ODbL) v1.0, and is available online at https://www.openstreetmap.org/.

### 2.1.3 GRanD

The GRanD version 1.3 (Lehner et al., 2011) contains 7250 records of reservoirs and their associated dams, whose attributes include names, spatial coordinates, surface area, storage capacity, dam height, construction year, main purpose, etc. GRanD, which with a high spatial accuracy and attribute coverage, is a highly-versatile geodatabase available to supporting regional or global analyses at a high spatial resolution, sophistication, and reliability (Lehner et al., 2011). Here, we take such data as

a part of the combined data. The GRanD is freely available for non-commercial use at http://sedac.ciesin.columbia.edu/pfs/grand.html.

### 2.2 Auxiliary Data

The Surface Water and Ocean Topography (SWOT) satellite could make the first global survey on the Earth's surface water, which was launched in December 2022, and would be used to continuously measure water surface elevation. The orbit data



of SWOT is used to calculate the number of orbits covered by each lake, so as to analyze its potential in the observation of
lakes. The mountain data (Korner et al., 2017) is used to overlap with the GLRSED dataset to distinguish the lakes located in
mountainous areas. The HydroBASINS (Lehner & Grill, 2013) was used to define basin boundaries, partition data for
processing and identify endorheic lakes. The GRanD, GlObal GeOreferenced Database of Dams (GOODD) (Mulligan et al.,
2020) and Georeferenced global Dams And Reservoirs (GeoDAR) database contain the information of 7250, 38667 and

24783 reservoirs/dams worldwide respectively, representing the vast majority of reservoirs on Earth. By combining
GLRSED with these three global reservoir databases, we partitioned human-managed reservoirs from natural lakes. The
SWOT River Database (SWORD) (Altenau et al., 2021), which contains a great deal of information on river topology and
network structures combined with multiple global river- and satellite-related datasets. Here, we attach the GLRSED dataset
to SWORD to distinguish lakes on rivers from independent lakes. The Randolph Glacier Inventory 6.0 (RGI 6.0), which

provides a global inventory of glacier polygons, was used to determine the distribution of glacier-fed lakes (Rignot et al.,
2014). In addition, the permafrost distribution data provided by the National Snow and Ice Data Center of the National
Aeronautics and Space Administration (NASA) was used to determine the permafrost-fed lakes. Table S1 in the
supplementary materials provides a summary of the sources of datasets above mentioned, together with their contributions to
the final products.

## 3 Method

Figure 1 shows the process flow. Firstly, the data was preprocessed. We downloaded the OSM data (OpenStreetMap
contributors, 2022) at December 2022 and then cleaned it by extracting lakes and reservoirs from all types of water.
Specifically, we extracted vectors containing "lake" or "reservoir" from the field of "name" and "fclass" in OSM. For China,
additional water bodies including "cuo" and "pond" were extracted. Due to limitations in the quality and format of OSM data,

we have not included all data. The scope of OSM data used in this paper is shown in Figure S2 in the supplementary
materials. The final processed OSM contains data on a total of approximately 0.85 million lakes. For HydroLAKES and
GRanD, the preprocessing process means using HydroBASINS to partition for subsequent processing.

Then, the integration processing of HydroLAKES, OSM and GRanD is carried out in different basins, together with a spatial
overlap with other auxiliary data and attribute settings.

For the spatial overlap with GeoDAR, we used a 270 m buffer to eliminate the impact of reservoir position deviations. The
same processing was performed for GOODD, except for buffering, as its high accuracy. We used the same method as
reference (Pi et al., 2022) to determine glacier-fed lakes by intersecting a spatially buffer zone of 1 km around the glacier
polygon obtained from the RGI 6.0t dataset. For the permafrost-fed lakes identification, there was a total of four permafrost
layers with varying degrees of coverage, and we identified the permafrost-fed lakes from low- to high-cover layers. That is,

when a lake belongs to both the low- and high-cover zones of the permafrost, it is assigned to the higher one. To identify the

endorheic lakes, we used the attribute "Endo" in a level-12 HydroBASINS dataset. The attribute of "Endo" variable that >0 is considered an endorheic basin. Therefore, all lakes that fell into these areas were considered endorheic lakes. Using the same method, we used SWORD to identify lakes located on rivers. Additionlly, by spatially joining our dataset with SWOT orbits, we calculated the number of orbits passing on each lake.

We labeled the above lake types as attributes of the dataset and calculated the area as well as shoreline attributes of each object through geographical calculations. Meanwhile, we preserved the ID attributes of HydroLAKES and GRanD, and for merging two or more, we only retained the first.

The above process was carried out using ArcGIS. Figure 2 shows a general scenario in which three datasets are integrated.

## 4 Results and Discussion

### 4.1 Patterns of the Distribution of Global Lakes/reservoirs Record in GLRSED

The final generated dataset contains 2.17 million lakes/reservoirs, with their spatial distribution shown in Figure 3. The distribution of the count and area of it by basin is shown in Figure 4. Table 2 shows the number of lakes in different area ranges on each continent and some countries, whose corresponding area can be found in Table S2 of the Supplementary Materials.

### 4.2 Attribute Table of the GLRSED

The attributes of GLRSED includes source lake data ID and numbers, name, area, length, types, etc. which are shown in Table 3, and the corresponding meanings and sources are also explained.

### 4.3 Coverage of SWOT Altimeter Satellite in GLRSED

SWOT (carry not only a nadir altimeter, but also a wide swath altimeter) provides water elevation measurements at the
intersections between satellite ground tracks and lakes. We calculated the SWOT orbits passing through each lake, as is shown in Figure 5. Near the North and South Poles, most lakes have at least 3 orbits passing through, which means that there can be at least 3 observations within a revisit cycle.

### 4.4 Permafrost-fed, Glacier-fed Lakes/reservoirs in GLRSED

In GLRSED, there are 964824 lakes/reservoirs located in the permafrost zone, including 963871 lakes and 953 reservoirs.
Among them, 123759 (538 for reservoirs) lakes are located in areas with a permafrost coverage of 0-10%, 133985 (188 for reservoirs) are located in areas with a permafrost coverage of 10-50%, while 148963 and 557164 (200 and 27 for reservoirs) are located in areas with a permafrost coverage of 50-90% and 90-100% respectively. Their spatial distribution is shown in Figure 6.

There are a total of 9371 lakes/reservoirs (total area: 14375 km2) located in glacial zones (Figure 7), which are mainly

distributed in Canada, Russia, United States, Greenland, Norway, Argentina, Chile and China, etc.

## 4.5 Lake/reservoirs Located in Mountainous Areas

Mountains supplying a substantial part of both natural and anthropogenic water demands, which are the water towers of the world and are highly-sensitive and prone to climate changes. Here, we identified 210612 lakes and 8926 reservoirs (see Figure 8) located in mountainous areas by intersecting GLRSED with mountains, which would help to study the impacts of

climate changes to the water towers.

## 4.6 Comparisons with HydroLAKES and OSM

In order to better understand the improvements and potential applications of this dataset, it was compared with two major datasets HydroLAKES (Messager et al., 2016) and OSM (OpenStreetMap contributors, 2022), which include global lakes/reservoirs. In terms of quantity, the GLRSED has 743813 and 1319625 more lakes/reservoirs than HydroLAKES and

lakes/reservoirs extracted from OSM, respectively. From a spatial perspective, the GLRSED has a same or larger spatial range than HydroLAKES and OSM, as it combines the spatial results of both, as shown in Figure 2. In terms of attributes, the HydroLAKES dataset records lake attributes, such as reservoir volume, lake depth, type, discharge, elevation of lake surface, etc. The attributes of OSM only include id, fclass and name. By integrating with multi-source data, like distribution data of river, dams, glaciers, permafrost, and high mountains, the GLRSED dataset not only preserves the basic information

of lakes, such as water area, shoreline length, data sources, etc., but also have more comprehensive attributes of lake types, such as endorheic lakes, reservoirs, glacier-fed lakes and permafrost-fed lakes, etc. While in HydroLAKES, only three lake types are included: lake, reservoir, lake control (i.e., natural lakes with regulating structures). Overall, our dataset may help researchers more conveniently conduct research related to lake types. We validated our data using remote sensing images at certain times (see Figure 9). In Figure 9a, HydroLAKES only records a small extent of lake. For the lake in Figure 9d,

HydroLAKES data has not recorded. Figures 9b and c show two cases where HydroLAKES and GRanD did not identify all lake surface water extent, while OSM identified more comprehensively. Although the acquisition date of remote sensing images and the vector drawing date may be different here, these examples demonstrate the general situation where single data are difficult to identify all lake water bodies. In most cases, our data has a large water body extent in space, which would avoid missing water bodies when using static lake surface water extent for initial recognition of lakes.

## 175  4.7 Uncertainties and limitations

Assessing the quality of the data is not easy as it depends on the data used, which have different levels of accuracy. For the HydroLAKES (Messager et al., 2016) database, which is the amalgamation of several sources, includes topographic and

remote sensing data, each with varying degrees of accuracy. For OSM data, its quality varies among different locations because it is created without any formal qualification.

In addition, there are certain errors may occur when crossing with third-party data in spatial. For example, when crossing the GLRSED with data of dams and reservoirs to obtain reservoir attributes, it is obtained through a certain distance buffer. The three reservoir datasets used (here is GRanD, GeoDAR and GOODD), although including most of the world's reservoirs, obviously, still have omissions. That is to say, a lake with a value of 0 in reservoir attributes may not necessarily be natural lake, but it may also be a reservoir. We attempted to annotate the source of each attribute as much as possible, so that users

could better use our dataset for analyses and expansions.

One limitation of the data is that it currently does not cover all lakes around the world, especially small and some occasional ones. Including more lakes/reservoirs and enriching their attributes (such as water volume, functions, etc.) is a direction for future. Despite the limitations and errors mentioned above, we believe that our data provides a lightweight basic reference data for global lake research, but further validation and adjustment are needed when studying specific topics.

**5 Conclusion**

In this paper, fully utilizing the value of datasets of HydroLAKES, OpenStreetMap and GRanD, we produced a global dataset on the geometric shape of lakes/reservoirs with 2.17 million individual features, called GLRSED. By spatially overlaying GLRSED with other auxiliary data, we identified mountain lakes, endorheic lakes, reservoirs, glacier-fed lakes and permafrost-fed lakes, etc. Within the scope of sizes and regions covered, these datasets are far more comprehensive than

existing ones.

Each uniquely-recognized lake/reservoir polygon in the GLRSED database has a set of morphometric attributes: area, perimeter, lake type, country and continent, etc., which would facilitate many studies, like the monitoring of Sustainable Development Goal 6. Combined with data in GLRSED, additional information could be added to support many analyses. For example, combining the object-oriented features and current attributes of the GLRSED database with the dynamic

capabilities of multi-temporal remote sensing images has the potential to monitor seasonal changes, such as identifying climate-change-sensitive areas and conducting research on the status of global lakes as well as the impact of climate changes.

**Data Availability**

1.Global Reservoir/Lake Surface Area Dataset (GLRSED) V1.0, Shapefile

The data be accessible from Zenodo: https://doi.org/10.5281/zenodo.8121174, under an Open Database License (ODbL) v1.0.



**Acknowledgments**

This work was supported by the International Research Center of Big Data for Sustainable Development Goals (CBAS2022GSP01), National Key R&D Program of China (No. 2022YFC3104900/2022YFC3104905).

**Author contribution**

Conceptualization, B.B., L.X. and Y.T.; methodology, software, validation, visualization, writing—original draft preparation, B.B. and L.X.; writing—review and editing, supervision, project administration, funding acquisition, G.C. and Y.T. All authors have read and agreed to the published version of the manuscript.

**Competing interests**

The authors declare no conflict of interest.

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





Table 1. List of recent existing datasets that contain surface water extent of lakes/reservoirs.

| Name and abbreviation | source | parameter | data format | scope | number | reference |
|---|---|---|---|---|---|---|
| Global Reservoir and dam Database (GRanD) | maps, national archives, the internet, etc. | surface extent, area, storage capacity, etc. | vector (shp) | global reservoirs | 6862 | (Lehner et al., 2011) |
| Joint Research Centre (JRC) Global surface water (GSW) | Landsat 5,7,8 | surface extent, area | raster | global surface water | NA | (Pekel et al., 2016) |
| HydroLAKES | Remote sensing, STRM, etc. | surface extent, area, volume, depth, etc. | vector (shp) | at least 0.1 km² lakes and reservoirs | 1.43 million | (Messager et al., 2016) |
| OpenStreetMap (OSM) | satellite and aerial imagery, etc. | surface extent | vector (shp, osm) | global surface water | NA | (OpenStreetMap contributors, 2022) |
| Global medium and small reservoirs | Landsat, Sentinel | surface extent, area | vector (shp) | >0.01 km² reservoirs, excluding 479 large ones | 71208 | (Donchyts et al., 2022) |
| RealSat | JRC GSW | surface extent, area | vector (shp) | > 0.1 km² lakes and reservoirs | 681137 | (Khandelwal et al., 2022) |

NA: not applicable.





**Table 2. The count distribution characteristics of lakes/reservoirs in GLRSED.**

| | | Count | | | | | | |
|---|---|---|---|---|---|---|---|---|
| | **Total** | **Area interval/km²** | | | | | | |
| | | **≤1** | **(1,10]** | **(10,100]** | **(100,200]** | **(200.500]** | **(500,1000]** | **> 1000** |
| **Continent** | | | | | | | | |
| **North America (NA)** | 1134504 | 1026445 | 99055 | 8202 | 413 | 253 | 68 | 68 |
| **Europe (EU)** | 377200 | 358891 | 16440 | 1682 | 81 | 63 | 18 | 25 |
| **Asia (AS)** | 382946 | 335471 | 42932 | 4048 | 229 | 165 | 54 | 47 |
| **South America (SA)** | 112522 | 102918 | 8479 | 955 | 75 | 50 | 24 | 21 |
| **Africa (AF)** | 84060 | 80922 | 2537 | 457 | 58 | 44 | 17 | 25 |
| **Oceania (OC)** | 80269 | 77082 | 2556 | 509 | 60 | 37 | 12 | 13 |
| **Countries with most lakes** | | | | | | | | |
| **Canada** | 887724 | 796985 | 83642 | 6505 | 295 | 193 | 53 | 51 |
| **USA** | 224339 | 210366 | 12500 | 1304 | 96 | 50 | 12 | 11 |
| **Russia** | 219679 | 184565 | 32857 | 2089 | 73 | 50 | 18 | 27 |
| **Australia** | 77117 | 74111 | 2419 | 474 | 54 | 35 | 11 | 13 |
| **Brazil** | 55739 | 52274 | 3031 | 342 | 38 | 31 | 14 | 9 |
| **Italy** | 54802 | 54623 | 157 | 16 | 4 | 2 | 0 | 0 |
| **China** | 50209 | 43372 | 5629 | 1036 | 87 | 57 | 19 | 9 |

**Table 3. Attribute table of GLRSED.**

| field | description and source |
|---|---|
| FID | This attribute is automatically assigned by ESRI. |
| Shape | This attribute is automatically assigned by ESRI. |
| Name | Name of the lake/reservoir which from HydroLAKES. |
| HydroLAKES_ID | ID of the corresponding features in the HydroLAKES, value null for no corresponding record. |
| GRanD_ID | ID of the corresponding features in the GRanD, value null for no |





| | corresponding record. |
|---|---|
| OSM_ID | ID of the corresponding features in the OSM, value null for no corresponding record. |
| Join_Cou_H | Number of HydroLAKES features included. |
| Join_Cou_O | Number of OSM features included. |
| Join_Cou_G | Number of GRanD features included. |
| Shore_len | Length of shoreline (i.e., polygon outline), in kilometers. |
| Lake_area | Lake surface area (i.e., polygon area), in square kilometers. |
| Longitude | Longitude of the lake pour point, in decimal degrees. |
| Latitude | Latitude of the lake pour point, in decimal degrees. |
| Country | Country that the lake (or reservoir) is located in. International or transboundary lakes are assigned to the country in which its corresponding lake pour point is located and may be arbitrary for pour points that fall on country boundaries. |
| Continent | Continent that the lake (or reservoir) is located in. Geographic continent: Africa, Asia, Europe, North America, South America, or Oceania (Australia and Pacific Islands). |
| Mountain | 0/1, lake located in mountainous areas or not. |
| Endorheic | 0/1, lake located in endorheic areas or not. |
| SWOT_obs | Number of SWOT ground tracks covered. |
| Reservoir | 0/1, reservoir identified based on GRanD, GOODD and GeoDAR or not. |
| SWORD | 0/1, lake located in river or not. |
| GOODD | 0/1, reservoir recorded by GOODD or not. |
| GeoDAR | 0/1, reservoir recorded by GeoDAR or not. |
| Glacier | 0/1, lake located in glacier areas or not. |
| Permafrost | 0-4, 1: lake located in areas with permafrost coverage of 0-10%; 2: lake located in areas with permafrost coverage of 10-50%; 3: lake located in areas with permafrost coverage of 50-90%; 4: lake located in areas with permafrost coverage of 90-100%; 0:lake not located in areas with permafrost coverage. |





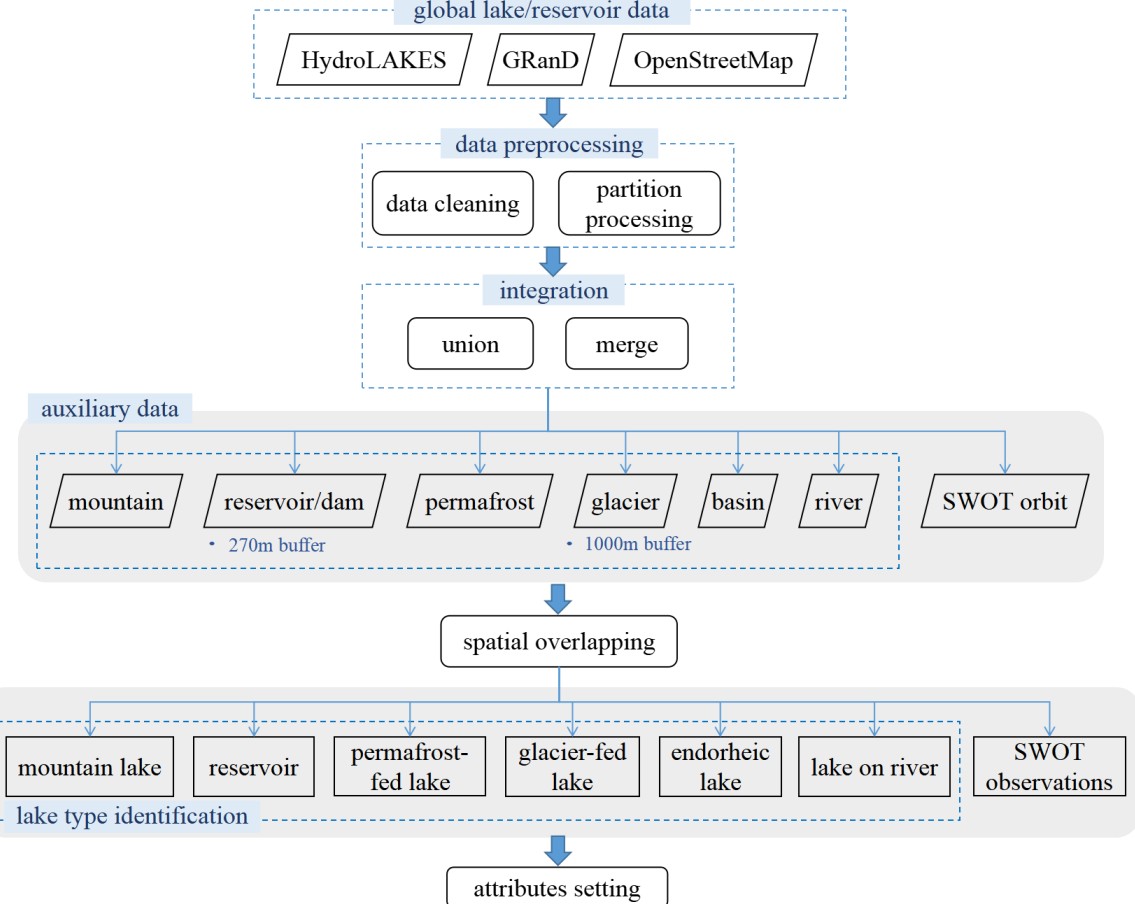

**Figure 1. Schematic flowchart of GLRSED production.**

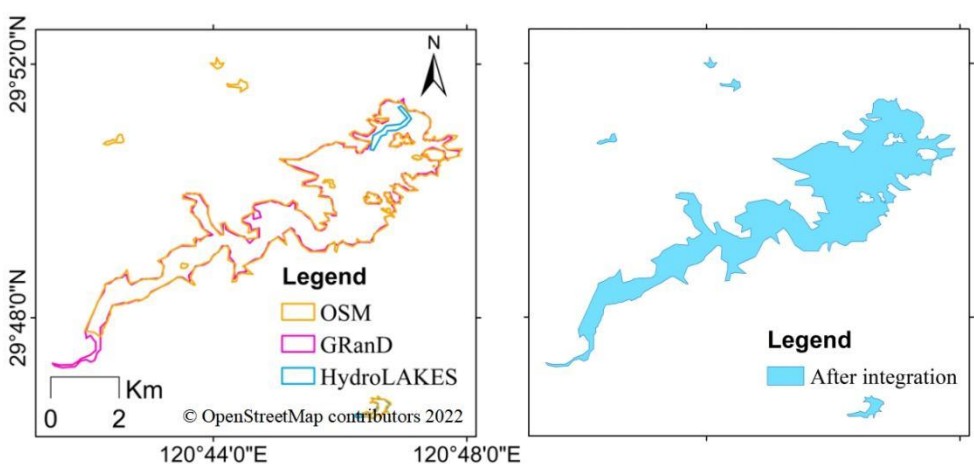

**Figure 2. An example of integrating HydroLAKES, OpenStreetMap (OSM) and GRanD. © OpenStreetMap contributors 2022. Distributed under the Open Data Commons Open Database License (ODbL) v1.0.**





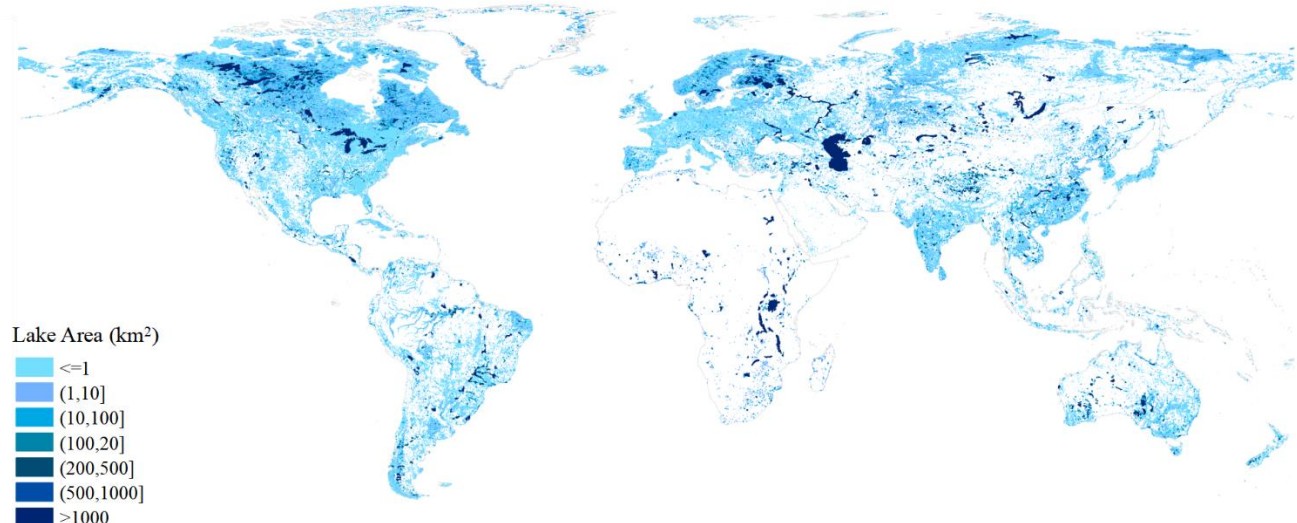

**Figure 3. Distribution of the lake/reservoirs in GLRSED.**





(a)

Lake Count (×10³)
- ≦ 1
- (1, 2]
- (2, 4]
- (4, 6]
- (6, 10]
- (10, 15]
- (15, 20]
- (20, 40]
- (40, 60]
- >60

(b)

Lake Area (×10³ km²)
- ≦ 1
- (1, 3.5]
- (3.5, 6.5]
- (6.5, 10]
- (10, 18]
- (18, 28]
- (28, 40]
- (40, 85]
- (85, 180]
- >180

**Figure 4. Distribution of (a) count and (b) area of the lakes/reservoirs in GLRSED by basin.**





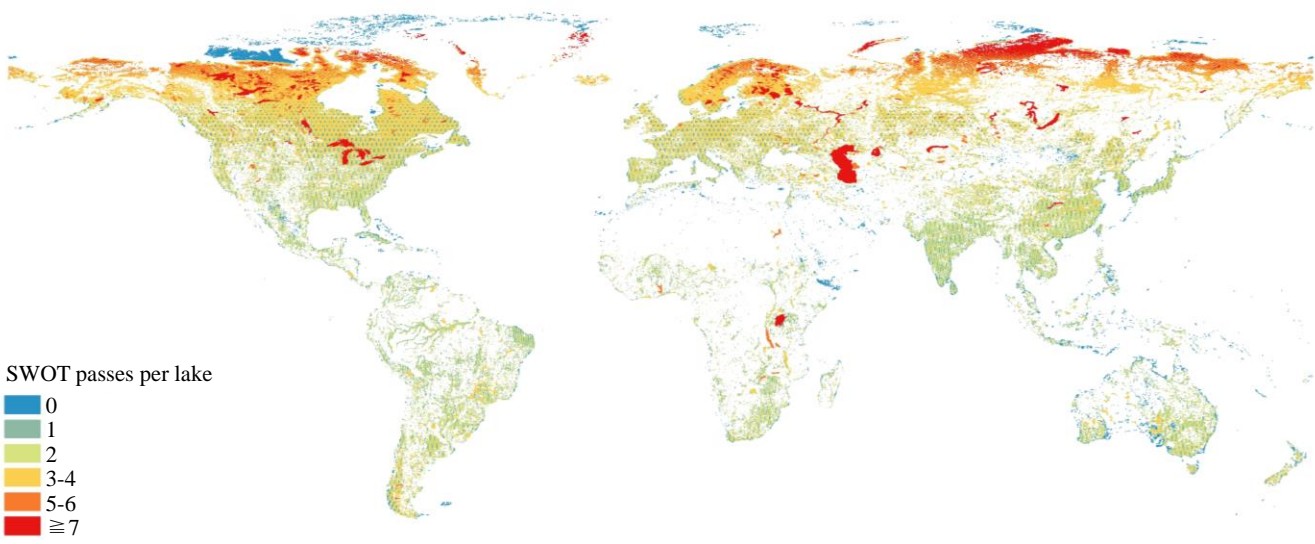

**Figure 5. SWOT ground tracks passing per lake/reservoir.**

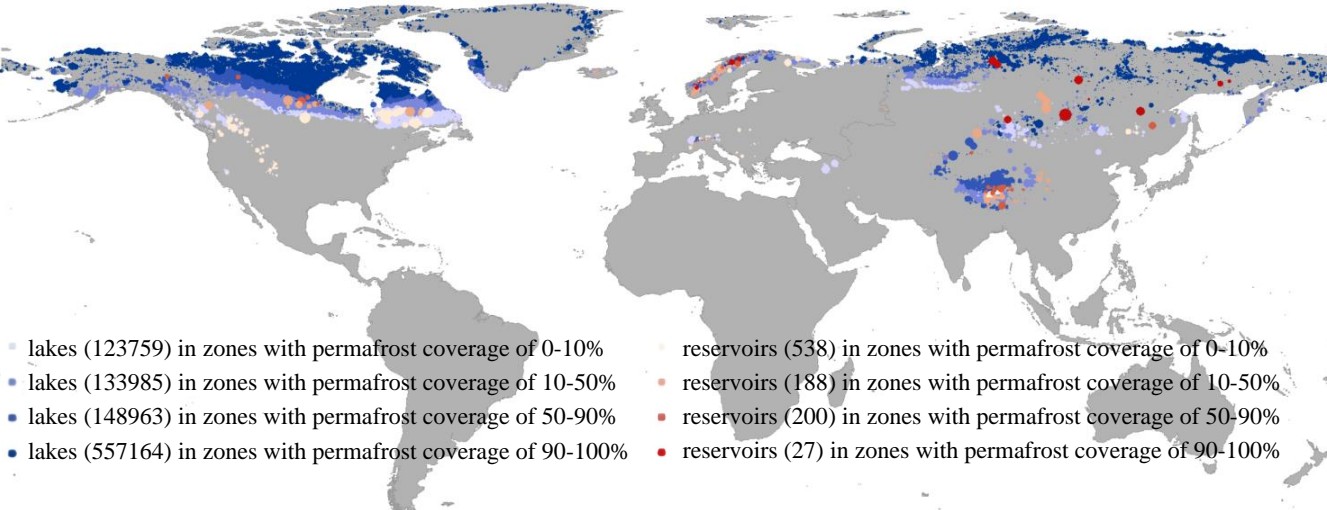

**Figure 6. Lakes (blue, count: 963871) and reservoirs (red, count: 953) located in permafrost zones in GLRSED (circle size represents the area).**



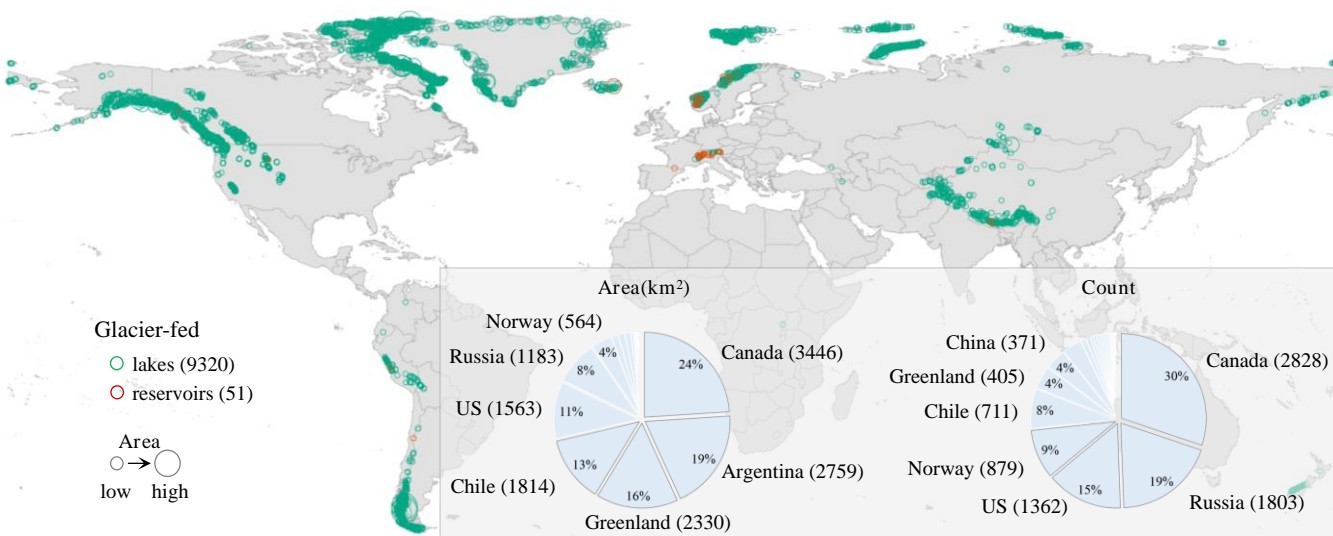

**Figure 7. Lakes (blue, count: 9320) and reservoirs (red, count: 51) located in glacier zones (circle size represents the area) in GLRSED.**

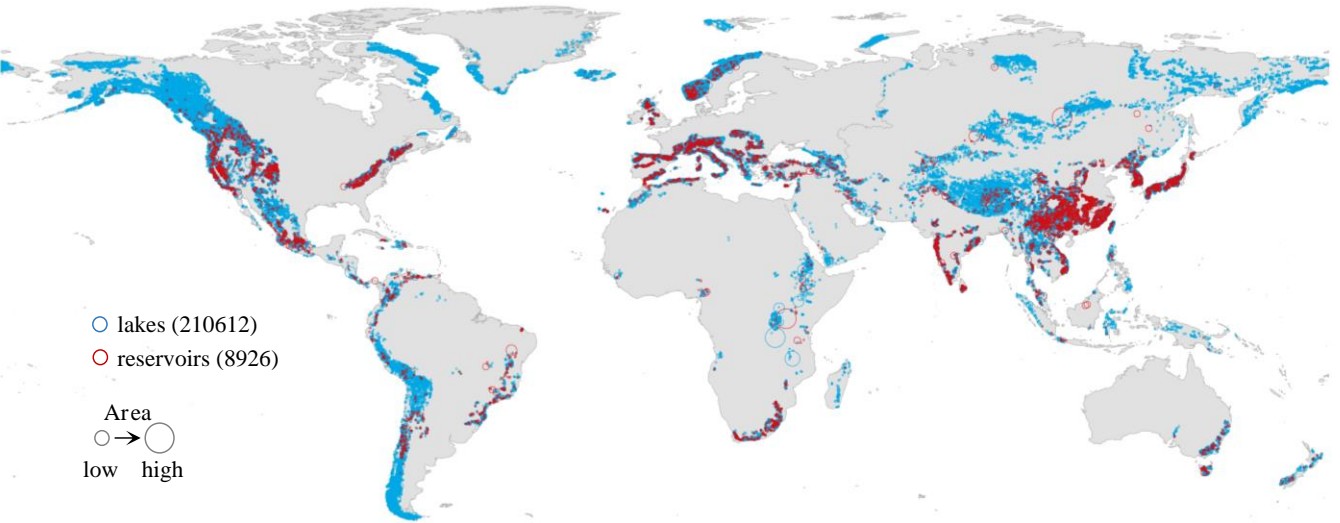


**Figure 8. Lakes (blue, count: 210612) and reservoirs (red, count: 8926) in mountainous zones in GLRSED. (circle size represents the area)**



**Figure 9. Examples of comparing** the **GLRSED dataset with HydroLAKES, OpenStreetMap (OSM) and GRanD data. (a) Tangpu Reservoir; (b) Danjiangkou Reservoir; (c) Dukan Reservoir; (d) Jilintai Reservoir. © OpenStreetMap contributors 2022. Distributed under the Open Data Commons Open Database License (ODbL) v1.0.**