# Peer review of "A Global Lake/Reservoir Surface Extent Dataset (GLRSED): An integration of HydroLAKES, GRanD and OpenStreetMap"

_Earth System Science Data, 2023_

## Author Comment (AC1)

**Review #1 of "A Global Lake/Reservoir Surface Extent Dataset (GLRSED): An integration of HydroLAKES, GRanD and OpenStreetMap"**

Dear reviewer,

We greatly thank you for the thorough review of the manuscript and the valuable comments. We have gone through these comments and suggestions carefully, and made revisions based on these comments and suggestions. Our responses are shown below.

AR: Author responses.

*"Italic"* represents the corresponding changes in the manuscript.

COMMENTS FROM REVIEWER

In this work the authors combine multiple hydrological datasets to improve the spatial representation of lakes and reservoirs as well as linking multiple important physical parameters to each water body. This work provide an important product for further inland water research and could specifically benefit from correction and integration with the upcoming SWAT measurements from the US National Aeronautics and Space Administration (NASA) and the French space agency Centre National d'Etudes Spatiales (CNES). I have some points with this manuscript which are listed hereunder, in general more details and clarity is required.

This work construct a Global Lake/Reservoir Surface Extent Dataset (GLRSED) by combining three main datasets (HydroLAKES, GRanD and OpenStreatMap) through a processing step followed by an integration of spatial extent and physical characteristics. Thereafter additional auxiliary data is derived through combination of GLRSED with additional spatial datasets (Mountains, permafrost, glaciers, endorheic catchments, etc.).

As it is now it is not clear how these steps were preformed. It can be inferred from the manuscript, that the spatial integration (spatial overlapping) was performed by finding the largest spatial extent for an arbitrary lake/reservoir present in the main datasets above. This process with each step need to be clearer described in the text. Furthermore Fig. 1 should be split into two figures and additional clarity and information should be provided. The first figure should show the exact working path used for constructing the GLRSED spatial dataset. Including the step where rivers was removed and how spatial interaction was treated, for example was a water body partly or fully removed when it was overlapped by a river segment? The second figure should show the exact derivation of lake type data from the auxiliary datasets including distances used (example the 1 km used for glacier interaction). The spatial overlapping clarification should also be present in the construction of the auxiliary datasets, it should be clear how lakes completely or partially within the set limit was classified.

AR: Thank you for your review and suggestions. We have made modifications to Figure 1 (see below) and divided it into two diagrams. The left shows spatial integration, the right shows attribute settings such as lake type, and the lower right shows the schematic diagram of the main spatial processing which provides clarification on some spatial overlap.

I'm a little confused about your question "including the step where rivers were removed and how spatial interaction was treated, for example was a water body partially or fully removed when it was overlapped by a river segment ?" In the spatial integration step, only the OSM have river data. But there is no situation of removing water body based on rivers in the processing of OSM. We obtain lakes/reservoirs by filter the fields of "name" and "fclass" of OSM. Whether a water body will be included depends on its description in the fields. If you mean the process of attribute setting. We determine whether a lake is independent or located on a river (intersecting or adjacent to the river) based on its spatial intersection situation with river, and there is no removal processing. Hope I answered your question.

"

[Figure]

*Figure 1. Schematic flowchart of GLRSED production."*

We have made some extensions to the method section in the manuscript, as follows:

*"3 Method*

*Figure 1 shows the process flow of GLRSED production. Firstly, the spatial integration. We downloaded the OSM data (OpenStreetMap contributors, 2022) at December 2022 and then cleaned it by extracting lakes and reservoirs from all types of water. Specifically, we extracted polygon features containing "lake" or "reservoir" from the fields of "name" and "fclass" in OSM. Considering the richness of local Chinese, in addition to "lake" and "reservoir", "cuo" and "pond" were used for China. This is for the purpose of collecting as much lakes/reservoirs as possible in this study. Due to limitations in the quality and format of OSM data, we have not included all data. The scope of OSM data used in this paper is shown in Figure S2 in the supplementary materials. The final processed OSM contains a total*

*of approximately 0.85 million lakes and reservoirs in this study. Then, the integration processing of HydroLAKES, OSM and GRanD is carried out in different basins. This mainly includes three steps: union (uniting multi-layers into one), merge (combine multiple selected features from the same layer into one) and explode multipart feature (separate a selected multipart feature into individual features). Finally, merge data from different basins. Figure 2 shows an example of integrating three datasets on a lake.*

*The second major step is the construction of attributes, which is mainly achieved by spatial overlap with a series of auxiliary data. Specifically, For the spatial overlap with GOODD and GeoDAR, we used a 270 m buffer to eliminate the impact of reservoir position deviations. For the glacier-fed lakes identification, we used the same method as references (Pi et al., 2022; Shugar et al., 2020), which intersects with a spatially buffer zone of 1 km around the glacier polygon obtained from the RGI 6.0t dataset. For the permafrost-fed lakes identification, the raster to polygon was first performed. As there are four permafrost layers with varying degrees of coverage, we assigned the permafrost-fed lakes by priority (from low- to high-cover layers), that is, when a lake belongs to both the low- and high-cover zones of the permafrost, it is assigned to the higher one (see Figure 1 for the spatial processing diagram). To identify the endorheic lakes, we used the field "ENDO" in level-12 HydroBASINS dataset. The field of "ENDO" that >0 is considered an endorheic basin. Therefore, all lakes that fell into these areas were considered endorheic lakes. The spatial processing of assigned by priority similar to that of the permafrost, is also performed when identify the endorheic lakes. By intersect our dataset with SWORD, we selected the features connected to the rivers. Using the same method, we identify lakes in mountains areas. By spatially joining our dataset with SWOT orbits, we calculated the number of orbits passing on each lake.*

*We calculated the area as well as shoreline attributes of each feature through geographical calculations. Meanwhile, we preserved the ID attributes of the three data source, lake depth attribute of HydroLAKES, and for merging two or more*

*features, we only retained the first one.*

*The above process was carried out using ArcGIS 10.8.*"

--Method section (in highlight version, same below)

As for the permafrost-, glacier fed lakes and endorheic lakes, the evidence provided in the manuscript is to general to tag these water bodies as such. In the text the authors classify a lake to be glacier fed if it is within (not clear exact how, see above) 1 km of glaciers, similar for endorheic and permafrost fed lakes. The evidence is enough to say these lakes/reservoirs are located in these regions, in fact the authors acknowledge this act in figure 6, but the details is not enough to classify how a lake or reservoir is fed.

AR: Thank you for your question. We used a similar method to literatures (Pi et al., 2022; Shugar et al., 2020) to determine glacial-fed lakes/reservoirs. As mentioned in these literature, the main focus of this method is lakes/reservoirs experiencing recent detachment from glaciers within a few decades or large supraglacial lakes that are highly distinguishable on long-term satellite observations. The meaning of fed here tends to say that lakes/reservoirs are located in these areas, rather than providing an exact feed source for a lake or reservoir. Explaining what exactly a lake or reservoir is fed by is not the focus of this study. When identifying the endorheic and permafrost -fed lake/reservoirs, we do not use 1 km buffer process.

I am missing lake/reservoir depth as a variable, which might be outside the scope of this study. But this work would benefit from the inclusion of this parameter. I leave it up to the authors to decide if they want to include this from the sources below and/or other works.

Choulga, M., Kourzeneva, E., Zakharova, E., and Doganovsky, A.: Estimation of the mean depth of boreal lakes for use in numerical weather prediction and climate modelling, Tellus A, 66, 21295, https://doi.org/10.3402/tellusa.v66.21295, 2014.

Lehner, B. and Döll, P.: Development and validation of a global database of lakes, reservoirs and wetlands, J. Hydrol., 296, 1–22, https://doi.org/10.1016/j.jhydrol.2004.03.028, 2004.

Toptunova, O., Choulga, M., and Kurzeneva, E.: Status and progress in global lake database developments, Adv. Sci. Res., 16, 57–61, https://doi.org/10.5194/asr-16-57-2019, 2019

AR: Thank you for your suggestions.

We have add lake depth attributes based on HydroLAKES. The corresponding field descriptions have been added in Table 3 as follow.

"

| | |
|---|---|
| *Depth_avg* | *Average lake depth (in meters), which is from HydroLAKES and defined as the ratio between total lake volume ('Vol_total') and lake area ('Lake_area') in HydroLAKES.* |

"

--Table 3

Due to the special format of GLDB data, we are still parsing its lake depth data to include it in our data. Whether it can be included depends on our future progress.

Short points:

Line 90: add reference for SWOT data (is it SWORD?) and the revisit cycle (21 days?).

AR: Thank you for your question. The SWOT here is not refer to SWOT Mission River Database (SWORD), it is refers to SWOT mission. We have added the access link of the data and revisit cycle, like below:

"*The 21 days of cycle orbits of SWOT (access link: https://www.aviso.altimetry.fr/en/missions/future-missions/swot/orbit.html, December 1, 2022), which are polygons containing SWOT track coverage for each pass, are*

*used to calculate the number of orbits covered by each lake, so as to analyze its*
*potential in the observation of lakes."*                       --Line 89-92

Line 91: The reader needs to know how a mountain is defined.

AR: Thank you for your suggestions. The mountain data used here is determined by ruggedness of terrain (Körner et al.,2011). This method make no distinction by elevation, but apply a minimum 200 m elevational amplitude among 3×3=9 grid points of 30" in 2.5' pixels. For a 2.5' pixel to be defined as mountainous, the difference between the lowest and highest of the 9 points must exceed 200 m. We made modifications in the manuscript.

*"The mountain (Körner et al., 2017) determined by ruggedness of terrain (Körner et al.,2011) is used to overlap with the GLRSED dataset to distinguish the lakes located in mountainous areas."*                     --Line 92-95

Line 98-99: Why did you use different search words in China and nowhere else? This should be uniform to avoid bias.

AR: Thank you for your question. This is a compromise between consistency of method and comprehensiveness of data. As this study is not focus on use consistent methods to obtain data, but to cover as many lakes as possible. In China, local languages are rich, making some description such as 'cuo' (Chinese pinyin) represent 'lake' (such as Namco on the Qinghai Tibet Plateau). We do not want to discard this data for consistency. This is similar to the issue with OSM data, which varies in richness due to the enthusiasm of volunteers and different policies in different regions. We explained this consideration in the manuscript.

*"Considering the richness of local Chinese, in addition to "lake" and "reservoir", "cuo" and "pond" were used for China. This is for the purpose of collecting as much lakes/reservoirs as possible in this study."*            --Line 111-113

Line 130 to 155: Combine and/or extend paragraphs

AR:Thank you for your suggestion. We have combined these paragraphs.

*"4.1 Patterns of the Distribution and Attribute of Global Lakes/reservoirs Record in GLRSED*

*..."* --Line 144-169

Fig. 1: redo as described

AR: We have made modifications as follow.

"

[Figure]

*Figure 1. Schematic flowchart of GLRSED production."*

Fig. 2: Add satellite image as in Fig. 9

AR:Thank you for your suggestion. We have added satellite image as follow.

"

[Figure]

*Figure 2. An example of integrating HydroLAKES, OpenStreetMap (OSM) and GRanD. © OpenStreetMap contributors 2022. Distributed under the Open Data Commons Open Database License (ODbL) v1.0."*

Fig. 3: Split into two frames, one with lakes and one with reservoirs

AR: We have split it into lakes and reservoirs, as follows.

"

[Figure]

*Figure 3. Distribution of the lakes (a) and reservoirs (b) in GLRSED."*

Fig. 4: Normalize lake count and area towards country area, and use the real spatial extent of all countries.

AR: The figure below shows the count and area of lakes/reservoirs by country. Due to data accuracy issues, we will delete the country field from our dataset and not display it in the manuscript. Users can use local data for statistical analysis at the national scale.

[Figure]

Figure. Distribution of (a) count and (b) area of the lakes/reservoirs in GLRSED by country.

Fig: 6: Simplify legend

AR: Thank you for your suggestions. We have simplified it as follow.

"

[Figure]

lakes
 · 123759 in 0-10%
 · 133985 in 10-50%
 · 148963 in 50-90%
 · 557164 in 90-100%

reservoirs
 · 538 in 0-10%
 · 188 in 10-50%
 · 200 in 50-90%
 · 27 in 90-100%

*Figure 6. Lakes (blue, count: 963871) and reservoirs (red, count: 953) of GLRSED (circle size represents the area) located in zones with different permafrost coverage."*

Table 2 and S2: Theses tables need more detail, it is hard to follow what they show. Furthermore make sure that Table 2 and Figure. 4a correspond to each other, it looks like Italy shouldn't be included in the table.

AR: Thank you for your suggestions. We have modified Tables 2. In terms of categories (i.e. count and area), Table 2 corresponds to Figures 4a and b, but Figure 4 further provides more detailed spatial statistical results at the basin scale.

*"Table 2. The count and area statistical in different size class of GLRSED by continent.*

| Continent | Count in different size class (km²) | | | | | | | |
|---|---|---|---|---|---|---|---|---|
| | Total | ≤1 | (1,10] | (10,100] | (100,200] | (200.500] | (500,1000] | >1000 |
| North America (NA) | 1134504 | 1026445 | 99055 | 8202 | 413 | 253 | 68 | 68 |
| Europe (EU) | 377200 | 358891 | 16440 | 1682 | 81 | 63 | 18 | 25 |
| Asia (AS) | 382946 | 335471 | 42932 | 4048 | 229 | 165 | 54 | 47 |
| South America (SA) | 112522 | 102918 | 8479 | 955 | 75 | 50 | 24 | 21 |
| Africa (AF) | 84060 | 80922 | 2537 | 457 | 58 | 44 | 17 | 25 |
| Oceania (OC) | 80269 | 77082 | 2556 | 509 | 60 | 37 | 12 | 13 |
| Continent | Area in different size class (km²) | | | | | | | |
| | Total | ≤1 | (1,10] | (10,100] | (100,200] | (200.500] | (500,1000] | >1000 |

| | | | | | | | |
|---|---|---|---|---|---|---|---|
| **North America (NA)** | 1338426.69 | 238318.9493 | 246445.154 | 201819.9311 | 56137.99471 | 77461.6361 | 46828.10755 | 471414.9173 |
| **Europe (EU)** | 236074.341 | 29880.49156 | 42873.6163 | 42705.93255 | 11050.32652 | 18373.02856 | 13247.87585 | 77943.06968 |
| **Asia (AS)** | 982963.7389 | 69294.78659 | 109627.7878 | 105224.5923 | 31043.28999 | 52152.34067 | 38198.0312 | 577422.9103 |
| **South America (SA)** | 155679.0442 | 15324.71931 | 22328.23708 | 26460.55369 | 10463.48472 | 15038.06657 | 15671.13724 | 50392.84561 |
| **Africa (AF)** | 277538.8052 | 4706.273394 | 7239.576697 | 13269.81673 | 8528.396997 | 13224.24828 | 11696.73701 | 218873.7561 |
| **Oceania (OC)** | 90132.09421 | 3754.826046 | 7457.528163 | 15024.83254 | 8549.869581 | 10336.48051 | 8849.141161 | 36159.41621 |

,,

Italy's high count of lakes/reservoirs here is due to its rich OSM data, which includes many small farmland reservoirs, as shown in the following figure (the red box represents OSM).

[Figure]

Thank you again for your review and suggestions! If you have any questions, please feel free to contact me.

Email: baibingxin@ouc.edu.cn

Yours sincerely,
Bingxin Bai